# Exponential Family Model-Based Reinforcement Learning via Score Matching

**Gene Li**
Toyota Technological
Institute at Chicago
gene@ttic.edu

**Junbo Li**
UC Santa Cruz
jli753@ucsc.edu

**Anmol Kabra**
Toyota Technological
Institute at Chicago
anmol@ttic.edu

**Nathan Srebro**
Toyota Technological
Institute at Chicago
nati@ttic.edu

**Zhaoran Wang**
Northwestern University
zhaoranwang@gmail.com

**Zhuoran Yang**
Yale University
zhuoran.yang@yale.edu

## Abstract

We propose an optimistic model-based algorithm, dubbed SMRL, for finite-horizon episodic reinforcement learning (RL) when the transition model is specified by exponential family distributions with $d$ parameters and the reward is bounded and known. SMRL uses score matching, an unnormalized density estimation technique that enables efficient estimation of the model parameter by ridge regression. Under standard regularity assumptions, SMRL achieves $\tilde{O}(d\sqrt{H^3T})$ online regret, where $H$ is the length of each episode and $T$ is the total number of interactions (ignoring polynomial dependence on structural scale parameters).

## 1 Introduction

This paper studies the regret minimization problem for finite horizon, episodic reinforcement learning (RL) with infinitely large state and action spaces. Empirically, RL has achieved success in diverse domains, even when the problem size (measured in the number of states and actions) explodes [35, 41, 28]. The key to developing sample-efficient algorithms is to leverage *function approximation*, enabling us to generalize across different state-action pairs. Much theoretical progress has been made towards understanding function approximation in RL. Existing theory typically requires strong linearity assumptions on transition dynamics [e.g., 51, 26, 8, 36] or action-value functions [e.g., 30, 52] of the Markov Decision Process (MDP). However, most real world problems are *nonlinear*. Our theoretical understanding of these settings remains limited. Thus, we ask the question:

*Can we design provably efficient RL algorithms in nonlinear environments?*

Recently, Chowdhury et al. [11] introduced a nonlinear setting where the state-transition measures are finitely parameterized exponential family models, and they proposed to estimate model parameters via maximum likelihood estimation (MLE). The exponential family is a well-studied and powerful statistical framework, so it is a natural model class to consider beyond linear models. Chowdhury et al. study exponential family transitions of the form:

$$\mathbb{P}_{W_0}(s'|s,a) = q(s') \exp\left(\langle \psi(s'), W_0 \phi(s,a) \rangle - Z_{sa}(W_0)\right), \qquad (1)$$

where $\psi \in \mathbb{R}^{d_\psi}$ and $\phi \in \mathbb{R}^{d_\phi}$ are known feature mappings, $q$ is a known base measure, $W_0$ is the unknown parameter to be learned, and $Z_{sa}$ is the log partition function that ensures the density integrates to 1. This transition model covers both linear dynamical systems as well as the nonlinear dynamical system (nonLDS), introduced by Mania et al. [33]. Linear dynamical systems with quadratic rewards, i.e., the linear quadratic regulator (LQR), have received much attention recently as

36th Conference on Neural Information Processing Systems (NeurIPS 2022).

an important testbench for RL in unknown, complex environments [19, 42, 27]. Thus, the work of Chowdhury et al. is a crucial step in bridging the gap between RL and continuous control.

However, MLE has several shortcomings. In order to estimate the parameter $W_0$ in (1), MLE requires estimating the log partition function $Z_{sa}$, which is computationally intensive. Practical implementations for MLE which estimate the log partition function via Markov Chain Monte Carlo (MCMC) methods can be slow and induce approximation errors [10]. These approximation errors can propagate in undesirable ways to the algorithm's planning procedure. Since the MLE $\hat{W}$ cannot be computed in closed form, Chowdhury et al. leave their estimator implicitly defined as solutions of the likelihood equations. As is typical for upper confidence RL (UCRL) algorithms, one constructs high probability confidence sets around the estimator. Due to the challenging modeling assumption, Chowdhury et al. employ confidence sets which are sums of KL divergences taken over the dataset.

In this work, we bypass these difficulties by instead proposing to learn the model parameters with *score matching*, an unnormalized density estimation technique introduced by Hyvärinen [22]. Score matching provides an explicit, easily computable closed form estimator for the model parameters by solving a certain ridge regression problem (Theorem 1). Moreover, we can employ high probability confidence sets which are ellipsoids centered at the estimator, a standard component in prior theoretical work on linear bandits and linear MDPs [e.g., 2, 26].

Our main results are as follows:

- We extend prior work on the score matching estimator in the i.i.d. setting by proving nonasymptotic concentration guarantees for non-i.i.d. data (Theorem 2).

- We consider regret minimization for the setting of exponential family transitions and bounded and known rewards. We design a model-based algorithm, dubbed SMRL, which achieves regret of $\tilde{O}(d\sqrt{H^3 T})$, with polynomial dependence on structural scale parameters (Theorem 3). Here, $d = d_\psi \times d_\phi$ is the total number of parameters of $W_0$, $H$ is the episode length, and $T$ is the total number of interactions. In each episode, SMRL uses score matching as a computationally efficient subroutine to estimate $W_0$ from data, then it constructs elliptic confidence regions around the estimator which contain $W_0$ w.h.p. and chooses policies optimistically based on such confidence regions. (This work assumes computational oracle access to an optimistic planner.)

Our regret guarantee matches that of Exp-UCRL, the model-based algorithm proposed by Chowdhury et al. When specialized to the nonLDS with bounded costs and features, score matching and MLE are equivalent estimators (Proposition 10). Here, the work of Kakade et al. [27] gives a tighter guarantee of $\tilde{O}(\sqrt{d_\phi(d_\phi + d_\psi + H)H^2 T})$; however we stress that our analysis applies to a broader class of models. Broadly speaking, we view score matching and MLE as complementary estimation techniques; while MLE relies on less assumptions, score matching enjoys computational efficiency and allows us to simplify both the algorithm and proofs. A detailed comparison is deferred to Section 5. In this work, we mainly compare against the papers [11, 27], but a broader summary of related work can be found in Appendix A.

**Notation.** For a vector $x \in \mathbb{R}^d$, we let $\|x\| := \|x\|_2$ denote the $\ell_2$ norm. For a matrix $M \in \mathbb{R}^{n \times d}$, we denote $\mathsf{vec}\,(M) \in \mathbb{R}^{nd}$ to be the vectorized version of $M$. For a matrix $M$, we also denote $\|M\|_2$ to be the operator norm and $\|M\|_F$ to be the Frobenius norm, i.e., $\|M\|_F := \|\mathsf{vec}\,(M)\|$. We also let $e_i \in \mathbb{R}^d$ and $E_{ij} \in \mathbb{R}^{n \times d}$ denote the canonical basis vectors and matrices respectively. For positive semidefinite matrices $A, B$, we let $A \preceq B$ to be $B - A \succeq 0$. For positive semidefinite matrix $A$ and vector $x$ we define $\|x\|_A := \sqrt{x^\top A x}$. For any $n \in \mathbb{N}$, we let $[n] := \{1, 2, \ldots, n\}$. For a twice differentiable function $f : \mathbb{R}^m \mapsto \mathbb{R}^n$ and any $i \in [m]$, we let

$$\partial_i f(x) := \left( \frac{\partial}{\partial x_i} f_1(x), \ldots, \frac{\partial}{\partial x_i} f_n(x) \right)^\top \in \mathbb{R}^n \text{ and } \partial_i^2 f(x) := \left( \frac{\partial^2}{\partial x_i^2} f_1(x), \ldots, \frac{\partial^2}{\partial x_i^2} f_n(x) \right)^\top \in \mathbb{R}^n.$$

We use the word "algorithm" liberally, since methods discussed in this paper as well as other papers require solving optimization procedures which can be computationally intractable.

## 2 Problem statement

We consider the setting of an episodic Markov Decision Process, denoted by $\mathrm{MDP}(\mathcal{S}, \mathcal{A}, H, \mathbb{P}, r)$, where $\mathcal{S}$ is the state space, $\mathcal{A}$ is the action space, $H \in \mathbb{N}$ is the horizon length of each episode, $\mathbb{P}$ is state transition probability measure, and $r : \mathcal{S} \times \mathcal{A} \mapsto \mathbb{R}$ is the reward function.

The agent interacts with the episodic MDP as follows. At the beginning of each episode, a state $s_1$ is chosen by an adversary and revealed to the agent. The agent picks a **policy function**, which is a collection of (possibly random) functions $\pi := \{\pi_h : \mathcal{S} \mapsto \Delta(\mathcal{A})\}_{h \in [H]}$ that determines the agent's strategy for interacting with the world. For each step $h \in [H]$, the agent observes the state $s_h$ and plays action $a_h \sim \pi_h(s_h)$. Afterwards, they observe reward $r_h(s_h, a_h)$, and the MDP evolves to a new state $s_{h+1} \sim \mathbb{P}(\cdot \mid s_h, a_h)$. The episode terminates at state $s_{H+1}$ after which the world resets.

The goal of the agent is to maximize their cumulative rewards through interactions with the MDP. Concretely, in our model-based setting the agent knows the reward function $r$ and that the transition model $\mathbb{P}$ lies in some model class $\mathcal{P}$, and they want to pick policies every episode to minimize **regret**, which we formally define later on.

Now we define the value function and action-value function. For every policy $\pi$, we can define a **value function** $V^\pi_{\mathbb{P},h} : \mathcal{S} \mapsto \mathbb{R}$, which is the expected value of the cumulative future rewards when the agent plays policy $\pi$ starting from state $s$ in step $h$, and the world transitions according to $\mathbb{P}$. In this paper, we include $\mathbb{P}$ in the subscript since we will analyze value functions for different models; if clear from context, we will drop the subscript $\mathbb{P}$. Specifically, we have:

$$V^\pi_{\mathbb{P},h}(s) := \mathbb{E}_{\mathbb{P}} \left[ \sum_{h'=h}^{H} r_{h'}(s_{h'}, a_{h'}) \;\middle|\; s_h = s, a_{h:H} \sim \pi \right], \quad \forall s \in \mathcal{S}, h \in [H].$$

Similarly, we define the **action-value** functions $Q^\pi_{\mathbb{P},h}(s,a) : \mathcal{S} \times \mathcal{A} \mapsto \mathbb{R}$ to be the expected value of cumulative rewards starting from a state-action pair in step $h$, following $\pi$ afterwards:

$$Q^\pi_{\mathbb{P},h}(s,a) := \mathbb{E}_{\mathbb{P}} \left[ \sum_{h'=h}^{H} r_{h'}(s_{h'}, a_{h'}) \;\middle|\; s_h = s, a_h = a, a_{h+1:H} \sim \pi \right], \quad \forall (s,a) \in \mathcal{S} \times \mathcal{A}, h \in [H].$$

An optimal policy $\pi^\star$ is defined to be the policy such that the corresponding value function $V^{\pi^\star}_{\mathbb{P},h}(s)$ is maximized at every state $s \in \mathcal{S}$ and step $h \in [H]$. Without loss of generality, it suffices to consider deterministic policies [48]. Given knowledge of the MDP $(\mathcal{S}, \mathcal{A}, H, \mathbb{P}, r)$, the optimal value function and action-value function can be computed via dynamic programming [47]; then the optimal policy can be computed as the policy that acts greedily with respect to the optional action-value function, i.e., $\pi^\star_h(s) = \arg\max_{a \in \mathcal{A}} Q^\star_{\mathbb{P},h}(s,a)$.

In the online setting, we will measure the performance of an agent interacting with the MDP over $K$ episodes via the notion of **regret**. In every episode $k \in [K]$, an adversary presents the agent with a state $s_1^k$, and the agent then chooses a policy $\pi^k$. The regret over $K$ episodes is the expected suboptimality of the agent's choice of policy $\pi^k$ compared to the optimal policy $\pi^\star$:

$$\mathcal{R}(K) := \sum_{k=1}^{K} \left( V_1^{\pi^\star}(s_1^k) - V_1^{\pi^k}(s_1^k) \right).$$

Implicit in the notation $\mathcal{R}(K)$ are the adversary's choice of initial states; our results for regret will hold for any sequence of adversarially chosen $\{s_1^k\}_{k \in [K]}$. We will also denote $T := KH$ as the total number of interactions the agent makes with the world.

## 2.1 Exponential family transitions

We consider the setting when the transition model class $\mathcal{P}$ is given by exponential family transitions and the the reward function $r : \mathcal{S} \times \mathcal{A} \mapsto \mathbb{R}$ is bounded a.s. in $[0,1]$ and known to the learner.[1]

**Definition 1** (Exponential family transitions, c.f., [11]). *Suppose $\mathcal{S} \subseteq \mathbb{R}^{d_s}$ and $\mathcal{A}$ is any arbitrary action set. Fix feature mappings $\psi : \mathcal{S} \mapsto \mathbb{R}^{d_\psi}$ and $\phi : \mathcal{S} \times \mathcal{A} \mapsto \mathbb{R}^{d_\phi}$, as well as base measure $q : \mathcal{S} \to \mathbb{R}$. For any matrix $W \in \mathbb{R}^{d_\psi \times d_\phi}$, let:*

$$\mathbb{P}_W(s'|s,a) := q(s') \exp\left( \langle \psi(s'), W\phi(s,a) \rangle - Z_{sa}(W) \right), \tag{2}$$

---

[1]Our results extend to settings where the rewards are not known but instead lie in some class $\mathcal{R} \subseteq (\mathcal{S} \times \mathcal{A} \to \mathbb{R})$ by including an additional reward estimation procedure in our algorithm; the regret would additionally depend on the complexity of $\mathcal{R}$.

*where $Z_{sa}(\cdot)$ is the log-partition function, which is completely determined once $\psi$, $\phi$, $q$, and $W$ are specified. Then we define the **exponential family transitions** model class $\mathcal{P}(\psi, \phi, q)$ as:*

$$\mathcal{P}(\psi, \phi, q) := \left\{ \mathbb{P}_W : \int_{\mathcal{S}} q(s') \exp(\langle \psi(s'), W\phi(s, a) \rangle) \, ds' < \infty, \ \forall (s, a) \in \mathcal{S} \times \mathcal{A} \right\}.$$

*Since $\psi, \phi, q$ are taken to be fixed and known to the learner, we will write the model class as $\mathcal{P}$.*

Along with this assumption, we introduce a notational convention. Given some real or vector-valued measurable function $f(s')$, we will write $\mathbb{E}_{sa}^W f(s')$ to denote the expected value of $f$ when $s'$ is drawn from the conditional distribution $\mathbb{P}_W(\cdot | s, a)$, i.e. $\mathbb{E}_{sa}^W f(s') := \int_{\mathcal{S}} f(s') \mathbb{P}_W(s' | s, a) ds'$.

Chowdhury et al. state their results for a setting where the unknown matrix $W_0 = \sum_{i=1}^{d} \theta_i A_i$, where the $A_i \in \mathbb{R}^{d_\psi \times d_\phi}$ are known matrices and $\theta \in \mathbb{R}^d$ is unknown. This setting can be viewed as a nonlinear analog of the linear mixture model considered in [36, 5]. Definition 1 is a special case with $d = d_\psi \times d_\phi$ and $A_{ij} := E_{ij}$. Our results can be extended to their general setting with minor modification. Quantitatively, we would replace factors of $d_\psi \times d_\phi$ with $d$ in both the concentration and regret guarantees, and similar to Chowdhury et al. we would introduce constants which depend on $A_i$. For simplicity of presentation, we study the fully unknown matrix setting.

## 2.2 Relationship to (non)linear dynamical systems

We now describe how Definition 1 generalizes the previously studied model class of (non)linear dynamical systems which have been explored in reinforcement learning and control theory literature.

First, we take a step back and describe linear dynamical systems (LDS), which govern the transition dynamics of the LQR problem.[2] An LDS is defined by the following transition dynamics:

$$s' = As + Ba + \varepsilon, \text{ where } \varepsilon \sim \mathcal{N}(0, \Sigma).$$

where $s, s' \in \mathbb{R}^{d_s}$, $a \in \mathbb{R}^{d_a}$, $A, B$ are appropriately sized parameter matrices, and $\Sigma \in \mathbb{R}^{d_s \times d_s}$ is a known covariance matrix. The problem of estimating $(A, B)$, known as *system identification*, has a long history (see Appendix A for more details).

Recently, system identification and regret minimization have been studied for nonlinear generalizations of LDS [33, 27]. In this paper, we refer to this setting as the *nonlinear dynamical system* (or nonLDS for short).[3] The nonLDS is described by the state transition model:

$$s' = W_0 \phi(s, a) + \varepsilon, \text{ where } \varepsilon \sim \mathcal{N}(0, \Sigma).$$

By setting $\phi(s, a) = [s, a]^\top$ and $W_0 = [A \ B]$, we recover the classical linear dynamical system. The nonLDS (and by extension the LDS) are special cases of Definition 1. This can be seen by writing out the pdf of the multivariate Gaussian distribution to get:

$$q(s') = \frac{1}{(2\pi)^{d_s/2} \det(\Sigma)^{1/2}} \exp\left( -\frac{\|s'\|_{\Sigma^{-1}}^2}{2} \right), \ \psi(s') = \Sigma^{-1} s', \ Z_{sa}(W_0) = \frac{\|W_0 \phi(s, a)\|_{\Sigma^{-1}}^2}{2}.$$

Lastly, note that Definition 1 is more general than that of the nonLDS, whose base measure $q(\cdot)$ and feature mapping $\psi(\cdot)$ must take a specific form given by the multivariate Gaussian. Definition 1 gives extra flexibility in the functions $q$, $\psi$, and $\phi$, which can be regarded as *design choices* for the practitioner. For example, one can pick the mapping $\psi$ the output of a neural network which captures the relevant features for the transition to $s'$; this is not permitted under the nonLDS setting.

## 3 Model estimation via score matching

In this section, we present the score matching method, the subroutine in our RL algorithm that estimates model parameters. We also introduce structural assumptions that enable us to derive a nonasymptotic concentration guarantee for the score matching estimator.

---

[2]Strictly speaking, our results do not handle unbounded costs, so they do not apply to the LQR problem.

[3]Kakade et al. [27] study kernelized version of this model, which they call the *kernelized nonlinear regulator*.

## 3.1 Background on score matching

Suppose we want to estimate the conditional density $\mathbb{P}(s'|s, a)$ of the form (2), given a dataset $\mathcal{D} = \{(s_t, a_t, s_{t+1})\}_{t\in[T]}$. MLE is the natural candidate for this estimation procedure, but it suffers from pitfalls. Solving for the MLE requires computing the log-partition function $Z_{sa}(\cdot)$. If the log-partition function is not known in closed form, it can be estimated using Markov Chain Monte Carlo methods [7, 10, 14]; however, this procedure may be computationally expensive. For some settings such as kernelized exponential families, MLE fails due to ill-posedness [21, 44].

Hyvärinen [22, 23] proposed score matching as an alternative to minimizing the log likelihood. Score matching minimizes the Fischer divergence, which is the expected squared distance between the score functions $\nabla_{s'} \log \mathbb{P}_W(s'|s, a)$. Specifically, we define the divergence between $\mathbb{P}_{W_0}$ and $\mathbb{P}_W$ for fixed $(s, a)$ as:

$$J(\mathbb{P}_{W_0}(\cdot|s, a)\|\mathbb{P}_W(\cdot|s, a)) := \frac{1}{2}\int_{\mathcal{S}} \mathbb{P}_{W_0}(s'|s, a) \left\|\nabla_{s'} \log \frac{\mathbb{P}_{W_0}(s'|s,a)}{\mathbb{P}_W(s'|s,a)}\right\|^2 ds'. \quad (3)$$

Before proceeding with the exposition of the score matching estimator, we list standard regularity conditions that are required for the analysis of score matching [cf., 44, 4].

**Assumption 1** (Regularity conditions)**.**

(A) $\mathcal{S}$ is a non-empty open subset of $\mathbb{R}^{d_s}$ with piecewise smooth boundary $\partial\mathcal{S} := \overline{\mathcal{S}} - \mathcal{S}$, where $\overline{\mathcal{S}}$ is the closure of $\mathcal{S}$.

(B) (Differentiability): $\psi(\cdot)$ is twice continuously differentiable on $\mathcal{S}$ with respect to each coordinate $i \in [d_s]$, and $\partial_i^j \psi(s)$ is continuously extensible to $\overline{\mathcal{S}}$ for all $j \in \{1, 2\}, i \in [d_s]$.

(C) (Boundary Condition): For all $(s, a) \in \mathcal{S} \times \mathcal{A}$ and $i \in [d_s]$, as $s' \to \partial\mathcal{S}$, we have:

$$\|\partial_i \psi(s')\| \, \mathbb{P}_{W_0}(s'|s, a) = o(\|s'\|^{1-d_s}).$$

(D) (Integrability): For all $i \in [d_s]$, $(s, a) \in \mathcal{S} \times \mathcal{A}$, let $p_{sa} := \mathbb{P}_{W_0}(\cdot|s, a)$. Then:

$$\|\partial_i \psi(s')\| \in L^2(\mathcal{S}, p_{sa}), \ \left\|\partial_i^2 \psi(s')\right\| \in L^1(\mathcal{S}, p_{sa}), \ \|\partial_i \psi(s')\| \, \partial_i \log q(s') \in L^1(\mathcal{S}, p_{sa}).$$

The key insight of Hyvärinen is that via an integration by parts trick, the divergence can be rewritten in a more amenable form. Essentially, these regularity conditions allow us to rewrite the conditional score function $J(W) := J(\mathbb{P}_{W_0}(\cdot|s, a)\|\mathbb{P}_W(\cdot|s, a))$ as:

$$J(W) = \frac{1}{2}\int_{\mathcal{S}} \mathbb{P}_{W_0}(s'|s, a) \cdot \sum_{i=1}^{d_s} \left[(\partial_i \log \mathbb{P}_W(s'|s, a))^2 + 2\partial_i^2 \log \mathbb{P}_W(s'|s, a)\right] ds' + C, \quad (4)$$

where $C$ does not depend on the parameter $W$. In Appendix B.1 we provide a more formal derivation of (4) for exponential family densities as well as further discussion on Assumption 1.

Crucially, (4) *can be estimated with samples without requiring computation of the partition function*, since the partition function vanishes when taking partial derivatives with respect to $s'$. This gives rise to the following formulation of the **empirical score matching loss** for a dataset $\mathcal{D} = \{(s_t, a_t, s_t')\}_{t\in[n]}$:

$$\hat{J}_n(W) := \frac{1}{2}\sum_{t=1}^{n}\sum_{i=1}^{d_s} \left((\partial_i \log \mathbb{P}_W(s_t'|s_t, a_t))^2 + 2\partial_i^2 \log \mathbb{P}_W(s_t'|s_t, a_t)\right). \quad \text{(SM-L)}$$

Furthermore, for any regularizer $\lambda > 0$, we can define the **empirical score matching estimator**:

$$\hat{W}_{n,\lambda} := \arg\min_{W} \hat{J}_n(W) + \tfrac{\lambda}{2} \|W\|_F^2. \quad \text{(SM-E)}$$

The following theorem gives a closed form expression for the empirical score matching estimator, when specialized to densities given by Definition 1.

**Theorem 1.** *For a dataset $\mathcal{D} = \{(s_t, a_t, s'_t)\}_{t \in [n]}$, (SM-L) can be written as:*

$$\hat{J}_n(W) = \frac{1}{2} \left\langle \text{vec}\,(W), \hat{V}_n \text{vec}\,(W) \right\rangle + \left\langle \text{vec}\,(W), \hat{b}_n \right\rangle + C,$$

where:

$$\hat{V}_n := \sum_{t=1}^{n} \sum_{i=1}^{d_s} \text{vec}\,\left( \partial_i \psi(s'_t) \phi(s_t, a_t)^\top \right) \text{vec}\,\left( \partial_i \psi(s'_t) \phi(s_t, a_t)^\top \right)^\top \in \mathbb{R}^{d_\psi d_\phi \times d_\psi d_\phi},$$

$$\hat{b}_n := \text{vec}\,\left( \sum_{t=1}^{n} \sum_{i=1}^{d_s} \left( \partial_i \log q(s'_t) \partial_i \psi(s'_t) + \partial_i^2 \psi(s'_t) \right) \phi(s_t, a_t)^\top \right) \in \mathbb{R}^{d_\psi d_\phi},$$

*and $C$ does not depend on $W$. In addition, (SM-E) can be computed as:*

$$\text{vec}\,\left( \hat{W}_{n,\lambda} \right) = -(\hat{V}_n + \lambda I)^{-1} \hat{b}_n. \tag{5}$$

Theorem 1 is a typical result in score matching literature, and can be derived as a corollary of Arbel and Gretton [4, Thm. 3]. For completeness, we give a proof in Appendix B.2.

For the rest of the paper, it is useful to derive matrix expressions for $\hat{V}_n$ and $\hat{b}_n$. We define the following functions:

$$\Phi(s, a) := [E_{11}\phi(s, a), E_{12}\phi(s, a), \dots E_{ij}\phi(s, a), \dots E_{d_\psi \cdot d_\phi}\phi(s, a)]^\top \in \mathbb{R}^{d_\psi d_\phi \times d_\psi},$$

$$C(s') := \sum_{i=1}^{d_s} \partial_i \psi(s') \partial_i \psi(s')^\top \in \mathbb{R}^{d_\psi \times d_\psi}, \quad \xi(s') := \sum_{i=1}^{d_s} \partial_i \log q(s') \partial_i \psi(s') + \partial_i^2 \psi(s') \in \mathbb{R}^{d_\psi}.$$

In addition, we use the subscript $t$ to denote the value of the above expressions on sample $(s_t, a_t, s'_t)$. We succinctly represent $\hat{V}_n = \sum_{t=1}^{n} \Phi_t C_t \Phi_t^\top$ and $\hat{b}_n = \sum_{t=1}^{n} \Phi_t \xi_t$.

**Computational efficiency.** We make a few remarks on the computation of the score matching estimator. From Theorem 1, we see that computing $\hat{W}_n$ does not require estimation of the log-partition function $Z_{sa}$. The objective is a *quadratic* function in $W$, which we can solve for via Equation (5). However, Equation (5) requires us to invert a $d_\phi d_\psi \times d_\phi d_\psi$ matrix, which takes time $O(d_\phi^3 d_\psi^3)$ and memory $O(d_\phi^2 d_\psi^2)$. This can be disappointing from a practical perspective, where the dimensionality of $\phi$ and $\psi$ can be large. Several additional considerations may remedy this. First, using the representer theorem, it is possible to show that $\hat{W}$ is the solution of a linear system of $n \cdot d_S$ variables, thus taking time $O(n^3 d_S^3)$ and space $O(n^2 d_S^2)$ [4, Thm. 1]. One can further reduce the dependence on $n$ using Nyström approximations [46]. Second, if we are in the structured setting where $W_0 = \sum_{i=1}^{d} \theta_i A_i$, where $\theta \in \mathbb{R}^d$ is unknown but the matrices $A_i \in \mathbb{R}^{d_\psi \times d_\phi}$ are known. Theorem 1 can be adapted to this setting, and solving for $\hat{\theta}_n$ will take time $O(d^3)$ and space $O(d^2)$.

### 3.2 Concentration guarantee

We provide concentration guarantees for score matching under some structural assumptions:

**Assumption 2** (Structural scaling)**.**

    **(A)** For any $(s, a) \in \mathcal{S} \times \mathcal{A}$ and $s' \sim \mathbb{P}_{W_0}(\cdot | s, a)$: we have $\xi(s')$ is $B_\psi$-subgaussian.

    **(B)** For any $(s, a) \in \mathcal{S} \times \mathcal{A}$ and $s' \sim \mathbb{P}_{W_0}(\cdot | s, a)$: we have $C(s')W_0\phi(s, a)$ is $B_c$-subgaussian.

    **(C)** For any $s' \in \mathcal{S}$: $\alpha_1 I \preceq C(s') \preceq \alpha_2 I$, where $\alpha_2 \geq \alpha_1 > 0$.

    **(D)** For any $(s, a) \in \mathcal{S} \times \mathcal{A}$: $\mathbb{E}_{sa}^{W_0} \psi(s') \psi(s')^\top - \mathbb{E}_{sa}^{W_0} \psi(s') \mathbb{E}_{sa}^{W_0} \psi(s')^\top \preceq \kappa I$.

The conditions in Assumption 2 are mostly adapted from prior work [44, 4, 11], with suitable modifications to accomodate our non-i.i.d. setting. Notably, Assumption 2 holds for nonLDS (when $\Sigma = \sigma^2 I$) with $B_\psi = \sigma^{-6}, B_c = 0, \alpha_1 = \alpha_2 = \sigma^{-4}$ and $\kappa = \sigma^{-2}$. Due to space considerations, we defer further discussion on Assumption 2 to Appendix B.3.

We can prove the following concentration guarantee.

**Theorem 2.** *Suppose Assumptions 1 and 2 hold. Let $\{\mathcal{F}_t\}_{t=1}^{\infty}$ be a filtration such that $(s_t, a_t)$ is $\mathcal{F}_t$ measurable, $s_t'$ is $\mathcal{F}_{t+1}$ measurable, and $s_t' \sim \mathbb{P}_{W_0}(\cdot|s_t, a_t)$.*

*For any $\delta \in (0, 1)$ and $\lambda > 0$, let:*

$$\beta_n := \sqrt{\frac{2(B_\psi + B_c)}{\alpha_1^2}} \cdot \sqrt{\log \frac{\det(\lambda^{-1}\hat{V}_n + I)^{1/2}}{\delta}} + \sqrt{\lambda} \|W_0\|_F .$$

*With probability at least $1 - \delta$, the score matching estimators of* (SM-E) *satisfy:*

$$\left\| \text{vec}\left(\hat{W}_{n,\lambda}\right) - \text{vec}(W_0) \right\|_{\hat{V}_n + \lambda I} \leq \beta_n, \quad \text{for all } n \in \mathbb{N}.$$

Theorem 2 is a *self-normalized* concentration guarantee, since the parameter error is rescaled by a data-dependent term $\hat{V}_n + \lambda I$. The proof is provided in Appendix B.4. The proof relies on the method of mixtures argument developed in the linear bandit literature [see, e.g., 2, 29].

## 4 Algorithm and main result

In this section, we present our main results, which introduce the Score Matching for RL (SMRL) algorithm (Algorithm 1) and provide regret guarantees.

### 4.1 Algorithm specification

Our algorithm works as follows. In each episode $k = 1, 2, \ldots, K$, we compute a elliptic confidence set $\mathcal{W}_k$ centered at our score matching estimator. In particular, we consider the $n := (k-1)H$ state transitions $\mathcal{D} = \{s_t, a_t, s_t'\}_{t=1}^n$ the agent has observed up until the beginning of episode $k$ and run the score matching estimator to get the prediction $\hat{W}_k := \arg\min_W \hat{J}(W) + \frac{\lambda}{2}\|W\|_F^2$, via (Equation (5)). In discussing our RL algorithm and its regret guarantees, we choose to index $\hat{W}$ and $\hat{V}$ by $k$ rather than $n$ to emphasize that these quantities are computed once per episode. We also drop the subscript $\lambda$ because it is fixed across the run of the algorithm.

Let $B_\star$ is some known upper bound on $\|W_0\|_F$. We define the confidence set

$$\mathcal{W}_k := \left\{ W \in \mathbb{R}^{d_\psi \times d_\phi} : \left\| \text{vec}\left(\hat{W}_k\right) - \text{vec}(W) \right\|_{\hat{V}_k + \lambda I} \leq \beta_k \right\}, \tag{6}$$

where

$$\beta_k := \sqrt{\frac{2(B_\psi + B_c)}{\alpha_1^2}} \cdot \sqrt{\log \frac{2\det(\lambda^{-1}\hat{V}_k + I)^{1/2}}{\delta}} + \sqrt{\lambda}B_\star.$$

Once the agent computes the confidence set $\mathcal{W}_k$, they observe a new state $s_1^k$ and compute an optimistic policy $\pi^k$ (line 5-6), which is the optimal policy with respect to the "best model" in $\mathcal{W}_k$. As long as $W_0 \in \mathcal{W}_k$, the optimistic planning procedure gives us an overestimate of the true value function $V_{\mathbb{P},1}^\star(s_1^k)$, ensuring sufficient exploration of the MDP. Lastly, the agent runs policy $\pi^k$ on the MDP to collect a new trajectory of data, which is added to the dataset $\mathcal{D}$.

### 4.2 Computational complexity

Algorithm 1 has two main components: model estimation (line 9) via score matching and optimistic planning (line 6). We have already discussed in Section 3 that the model estimation can be computed efficiently. Planning is a different story. Even planning with a *known model*, i.e., solving the problem $\pi^k = \arg\max_\pi V_{\mathbb{P}_W,1}^\pi(s_1^k)$, is already challenging without imposing further structure. However, it can be approximated with model predictive control [34, 49]. Furthermore, even with access to a planning oracle, *optimistic planning* is known to be NP-hard in the worst case [15]. In this work, we assume computational oracle access to the optimistic planner that solves (line 6) and leave developing efficient approximation algorithms to future work. One alternative to optimistic planning is to employ posterior sampling methods in conjunction with (approximate) planning oracles; the Bayesian regret can be theoretically analyzed using well-established techniques [e.g., 37, 11].

---

**Algorithm 1** Score Matching for RL (SMRL)

---
1: **Input:** Regularizer $\lambda$ and constants $B_\psi, B_c, B_\star, \kappa, \alpha_1$.
2: **Initialize:** starting confidence set $\mathcal{W}_1 = \mathbb{R}^{d_\psi \times d_\phi}$, confidence widths $\{\beta_k\}_{k \geq 1}$, dataset $\mathcal{D} = \emptyset$.
3: **for** episode $k = 1, 2, 3, \cdots, K$ **do**
4:     **Planning:**
5:     Observe initial state $s_1^k$
6:     Choose the optimistic policy: $\pi^k = \arg\max_\pi \max_{W \in \mathcal{W}_k} V^\pi_{\mathbb{P}_W, 1}(s_1^k)$
7:     **Execution:**
8:     Execute $\pi^k$ to get a trajectory $\{s_h^k, a_h^k, r_h^k, s_{h+1}^k\}_{h \in [H]}$, and add it to $\mathcal{D}$.
9:     **Solve for score matching estimator $\hat{W}_k = \arg\min_W \hat{J}(W) + \frac{\lambda}{2} \|W\|_F^2$ via (5)**
10:     **Update confidence set $\mathcal{W}_{k+1}$ via (6)**

---

## 4.3 Regret guarantee

We now provide our main result, which is a $\sqrt{T}$-regret guarantee on the performance of SMRL.

**Theorem 3** (SMRL Regret Guarantee). *Suppose Assumptions 1 and 2 hold. Set $\lambda := 1/B_\star^2$ and fix $\delta \in (0, 1)$. Then with probability at least $1 - \delta$:*

$$\mathcal{R}(K) \leq C \sqrt{\gamma_{K+1} \cdot \left( \frac{\kappa(B_\psi + B_c)}{\alpha_1^3} \left( \gamma_{K+1} + \log 1/\delta \right) + \frac{\kappa}{\alpha_1} + H \right)} \cdot \sqrt{H^2 T},$$

*where $C > 0$ is an absolute constant and $\gamma_{K+1} := \log \det(\lambda^{-1} \hat{V}_{K+1} + I)$. If $\|\phi(s, a)\| \leq B_\phi$ for all $(s, a)$, then $\mathcal{R}(K) \leq \tilde{O}(d_\psi d_\phi \cdot \sqrt{H^3 T})$, where the $\tilde{O}$ hides log factors and $\mathrm{poly}(\kappa, B_\psi, B_c, \alpha_1^{-1})$.*

The proof is presented in Appendix C. A few remarks are in order. Our regret guarantee depends on the number of model parameters $d_\psi \cdot d_\phi$ and not on the state and action space sizes, thus making our algorithm sample-efficient in large-scale environments where $|\mathcal{S}|$ and $|\mathcal{A}|$ are infinite. Additionally, it is easy to redo the analysis when the parameter matrix is structured, i.e., $W_0 = \sum_{i=1}^d \theta_i A_i$, to see that the regret guarantee depends on $d$ instead of $d_\psi \times d_\phi$. Thus, we can recover the same regret guarantee of $\tilde{O}(d\sqrt{H^3 T})$ that Chowdhury et al. provide.

On the more technical side, in Theorem 3, we require $\phi$ to be a bounded feature mapping, which linear dynamical systems do not satisfy in general (recall $\phi = [s, a]^\top$, and $s, a$ can have unbounded norm). We need this to provide a bound on a certain "information gain" quantity $\gamma_k = \log \det(\lambda^{-1} \hat{V}_k + I)$ [cf., 43, 27]; however, the bounded $\phi$ assumption can be substantially weakened because our proof only requires $\sum_{h=1}^H \|\phi_h\|^2$ to be bounded in every episode with high probability. In particular, if one restricts to controllable policies which do not blow up norm of the state [e.g., 13], then the information gain term can be bounded.

## 5 Score matching vs maximum likelihood estimation

In this section, we provide a detailed comparison of score matching with maximum likelihood approaches. First we compare for exponential family transitions of Definition 1; then we specialize our comparison for the nonLDS setting. Lastly, we provide numerical evidence to demonstrate a setting where (a variant of) SMRL is superior.

### 5.1 General comparison for exponential family transitions

Score matching and MLE can be viewed as complementary techniques for density estimation; we highlight the relative pros and cons of SMRL vs Exp-UCRL.

In general, Exp-UCRL can be applied to more settings than score matching, due to the fact that score matching requires regularity conditions (Assumption 1) that are needed for the derivation of (4). In particular, we require $\mathcal{S}$ to be a Euclidean space and the feature vector $\psi : \mathcal{S} \to \mathbb{R}^{d_\psi}$ to be a twice-differentiable mapping. In this sense, the scope of SMRL is more limited than that of Exp-UCRL. For example, while tabular and factored MDPs can be modeled as exponential family

transitions via the softmax parameterization,[4] we cannot prove regret guarantees for SMRL due to the differentiability requirement. Since the MLE estimator of Chowdhury et al. can be computed in $\text{poly}(S, A)$ time, in the tabular and factored MDP settings we would prefer to run Exp-UCRL.

Among models given by Definition 1 where *both* score matching and MLE can be applied, score matching is preferred because the estimator can be computed in closed form as the solution to a ridge regression problem, and elliptic confidence sets can be constructed around it using Theorem 2. For the MLE, this is not possible in general. Chowdhury et al. implicitly define the estimator as the solution to the likelihood equations, and their confidence set is constructed in a complicated fashion, in terms of sums of KL divergences taken over the dataset. Thus, while we are unable to claim overall computational tractability of Algorithm 1 due to the computational difficulty of optimistic planning, score matching enables us to estimate model parameters efficiently, an improvement from Exp-UCRL.

We now compare the regret guarantee of Theorem 3 with previous results; the detailed calculations are deferred to Appendix D.1. We achieve the same order-wise guarantee as Chowdhury et al.(Thm. 2) of $\tilde{O}(d_\phi d_\psi \cdot \sqrt{H^3 T})$. In terms of problem constants, both bounds depend on $\sqrt{\kappa}$, but we (1) require the constants $B_\psi$ and $B_c$, (2) replace dependence on strict convexity of the log partition function with the parameter $\alpha_1$.

## 5.2 Comparison with prior work for nonLDS

Now we compare our results for SMRL with the results for Exp-UCRL (Chowdhury et al.) and LC³ (Kakade et al.) for the nonLDS problem with bounded and known rewards. For simplicity we will take the transition noise to be $\mathcal{N}(0, \sigma^2 I_{d_s})$. We will also assume that $\|W_0\|_F \leq B_\star$ and that the feature vectors are bounded as $\|\phi(s, a)\| \leq B_\phi$ for all $(s, a) \in \mathcal{S} \times \mathcal{A}$. All three are similar UCRL-style algorithms, and we compare the parameter estimation, confidence sets, and regret guarantee.

**Estimation and confidence set construction.** For nonLDS, score matching and MLE are equivalent estimators (see Proposition 10 for a formal statement). Thus, in all three algorithms, the parameter estimation *procedure* is identical, up to rescaling of regularization parameter $\lambda$. To further facilitate comparison, we will hereafter fix the $\lambda$ of each algorithm such that the parameter estimation is the same as LC³ (for any fixed dataset). Our choices are detailed in Appendix D.2.

Once we have fixed the parameter $\lambda$ for each algorithm, the main distinction lies in the confidence set construction. While all three algorithms essentially utilize the same optimistic planning procedure, optimistic planning depends on the confidence sets constructed in each episode. The chosen policies and the resulting trajectories will be different in all three algorithms. The confidence sets constructed for each paper are essentially the tightest self-normalized bound one can prove, so it is hard to directly compare the confidence sets from paper to paper due to the difference in analyses. Generally speaking, SMRL uses Frobenius norm bounds (Theorem 2), Exp-UCRL uses a mixture of both Frobenius and spectral [11, Sec. 3.1], and LC³ uses only spectral norm bounds [27, Eq. 3.2].

**Regret guarantee.** In terms of the regret guarantee, Theorem 3 gives us a regret guarantee of $\tilde{O}\big(\sqrt{d_\phi d_\psi \cdot (\sigma^4 d_\phi d_\psi + H)H^2 T}\big)$, while a bound of $\tilde{O}\big(\sqrt{d_\phi^2 d_\psi^2 (1 + \sigma^{-2} B_\star^2 B_\phi^2 H)H^2 T}\big)$ can be derived for Exp-UCRL. Note that the latter bound depends polynomially on the scale of $W_0$ and $\phi$. Kakade et al. (Remark 3.5) give a bound for LC³ of $\tilde{O}(\sqrt{d_\phi(d_\phi + d_\psi + H)H^2 T})$, without polynomial dependence on $\sigma^2$ and the scale of $W_0$ and $\phi$. We conjecture that the $\sigma^2$ dependence is an artifact of our analysis, but it is less clear whether the dependence on $d_\phi, d_\psi$ can be improved.

## 5.3 Experiments on synthetic MDP

We demonstrate end-to-end benefits of using score matching in a (highly stylized) synthetic MDP; see Figure 1. In our constructed MDP, the transition function is multimodal; the action choice affects the location of the modes of the next state density. The reward is constructed so that $a = +1$ leads to higher reward than $a = -1$ at most states. To enable fair comparison, we *fix* a simple random sampling shooting planner [39] and evaluate three model estimation procedures: score matching with

---

[4]There is a mild technical issue, since Definition 1 cannot capture transitions with probability 0, so we must assume that the support of the transitions is known in advance. See the paper [11] for more details.

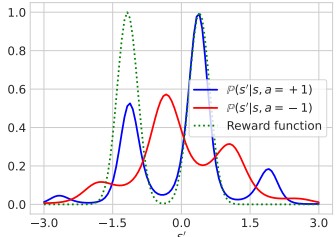 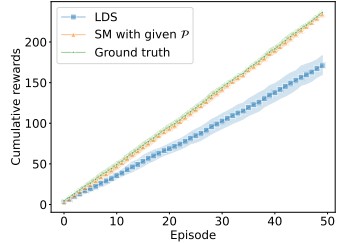 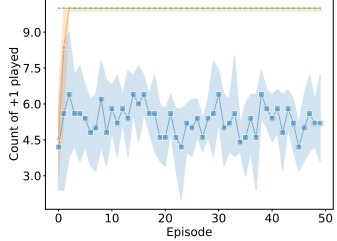

(a) MDP transition and reward      (b) Cumulative rewards      (c) Planner action choices

Figure 1: Comparing SM vs fitting an LDS for a synthetic MDP, with $\mathcal{S} = \mathbb{R}$, $\mathcal{A} = \{+1, -1\}$, $H = 10$, initial state distribution $\mathrm{Unif}([-1, +1])$, $\mathbb{P}(s'|s, a) = \exp(-s'^{1.7}/1.7) \cdot \exp(\sin(4s')(s + a))$, and $r(s, a) = \exp(-10(s - \pi/8)^2) + \exp(-10(s + 3\pi/8)^2)$. (a) plots $\mathbb{P}$ for a single starting state $s = 0.5$ for $a = +1$ and $a = -1$; the reward $r$ is superimposed. Taking $a = +1$ is more likely to transition to states with high reward. (b) plots cumulative reward for fixed planner with varying model estimation: SM with the given $\mathcal{P}$, fitting an LDS, and a baseline with the ground truth model. (c) plots the number of steps in every episode where $a = +1$ is picked by the planner. In (b) and (c), shaded areas correspond to 95% confidence intervals.

the given class $\mathcal{P}$, fitting an LDS via MLE, and a baseline where planner is supplied the ground truth $\mathbb{P}$. (For this simple one-dimensional RL task, one can also numerically compute the MLE with the given $\mathcal{P}$. However, this approach does not scale to RL tasks with high-dimensional states.) Fitting an LDS does poorly because the LDS density is not expressive enough to differentiate between $a = +1$ and $a = -1$, while score matching estimates the density well, so the planner quickly learns to pick $a = +1$. Our experiments suggest that modeling the transition $\mathbb{P}$ via the richer Definition 1 can yield end-to-end benefits for RL tasks. Further experimental details can be found in Appendix F.

# 6 Conclusion

In this paper, we show $\sqrt{T}$-regret guarantees for a reinforcement learning setting when the state transition model is an exponential family model, a challenging nonlinear setting. Under this modeling assumption, the commonly employed MLE may be intractable; we bypass such issues by proposing to learn the model via the score matching method.

We conclude with a few possible directions for future work.

- *Model Misspecification:* Proving guarantees for SMRL when the underlying transition $\mathbb{P}$ do not lie in the model class $\mathcal{P}$ but instead is well-approximated by $\tilde{\mathbb{P}} \in \mathcal{P}$ is an interesting direction.

- *Arbitrary State Spaces:* A key limitation of the score matching estimator is that it requires that the state space $\mathcal{S}$ must be a subset of the Euclidean space $\mathbb{R}^{d_s}$ and the feature mapping $\psi$ to be twice differentiable; therefore it cannot handle arbitrary state spaces. One important direction is extending the score matching algorithm to discrete state spaces such as tabular/factored MDPs through a suitable modification of the estimation procedure [e.g., 23, 31].

- *Kernelization:* We would like to extend our guarantees to the *kernel conditional exponential family* (KCEF) setting of Arbel and Gretton [4], i.e., when the conditional model is $\mathbb{P}_f(s'|s, a) := q(s') \cdot \exp\left(\langle f, \Gamma_{sa} k(s', \cdot)\rangle - Z_{sa}(f)\right)$, where $f$ lies in some vector valued Reproducing Kernel Hilbert Space (RKHS) $\mathcal{H}$, $k(s', \cdot)$ lies in an RKHS $\mathcal{H}_S$, and $\Gamma_{sa} : \mathcal{H}_S \to \mathcal{H}$ is an operator that depends on $(s, a)$. This generalizes our finite dimensional setting and is a special case of the conditional family where the inner product is $\langle f, \phi(s, a, s')\rangle$, studied by Canu and Smola [9]. In the KCEF setting, MLE becomes computationally intractable; yet score matching can be kernelized and efficiently computed, and fast approximation methods exist [46]. Our theory does not hold for the KCEF because the parameter $\alpha_1 = 0$. Instead, one might be able to adapt the range-space assumption from the paper [4] to the non-i.i.d. setting.

## Acknowledgments and Disclosure of Funding

This work is supported by funding from the Institute for Data, Econometrics, Algorithms, and Learning (IDEAL). We thank Pritish Kamath, Danica J. Sutherland, Akshay Krishnamurthy, and Wen Sun for helpful discussions. Part of this work was done while GL, ZW, and ZY were participating in the Simons Program on the Theoretical Foundations of Reinforcement Learning in Fall 2020.

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
