# OpenReview forum: "Exponential Family Model-Based Reinforcement Learning via Score Matching"
_NeurIPS.cc/2022/Conference — NeurIPS 2022 Accept_

### Official Review · Reviewer_QcxS · 2022-07-06

**Rating:** 7
**Confidence:** 4
**Soundness:** 4 excellent
**Presentation:** 4 excellent
**Contribution:** 3 good

**Summary:**

The paper proposes a novel method for estimating the transition model of an MDP parametrized by exponential functions, in the episodic finite-horizon model-based RL setting with bounded and known rewards. The method uses score matching that reduces to ridge regression of the known parameters of the exponential parametrization, thus eliminating the difficulties associated with estimating the partition function in MLE-based methods. An estimate of the online regret for the new algorithm is derived, too.

**Questions:**

What are the actually achievable advantages of the proposed method in comparison with a reasonable baseline, for example MLE? Section 5 provides a comparison in general terms, but there is no empirical verification. Maybe such a verification on prototypical test MDPs would be useful in illustrating the analysis?

**Limitations:**

The authors have addressed limitations adequately, for example they have conscientiously pointed out that their analysis does not cover LQR problems, due to their unbounded costs. I do not see any potential negative societal impacts of this work.

Minor typos:
P.4 L.136,138: "gaussian" -> "Gaussian"

**Strengths And Weaknesses:**

The main contribution of the paper is probably the idea to apply score matching to the exponential parametrization proposed earlier, resulting in a more efficient estimation algorithm. This is a very non-obvious and original advance, and given that the investigated parametrization subsumes a very large class of systems encountered in practice, the practical significance of this advance is likely to be high. However, the paper is entirely theoretical, and it is difficult to understand the computational advantages of the proposed method without at least some kind of empirical evaluation.

---

> ### Author Response · Authors · 2022-08-02
> **Thank you for your review.**
>
> We thank the reviewer for their comments and time, and have no corrections or objections.
>
> *To address the empirical performance*, we refer the reviewer to our response to Reviewer rkgm (so as not to be repetitive).

---

> > ### Comment · Reviewer_QcxS · 2022-08-08
> > **Maintaining evaluation**
> >
> > After reading the other reviewers' comments, I will keep my score unchanged.

---

### Official Review · Reviewer_LwWm · 2022-07-10

**Rating:** 6
**Confidence:** 3
**Soundness:** 3 good
**Presentation:** 3 good
**Contribution:** 2 fair

**Summary:**

This paper studies model-based reinforcement learning for episodic MDP whose transition model is parametrized by exponential families with features of state and action.
To estimate the model parameter, the author uses the score matching technique to minimize the expected squared distance between the score functions.
To promote exploration, the author utilizes the optimistic planning.
The author states that under some regularity assumptions, the suggested algorithm achieves $\tilde{\mathcal{O}}(d \sqrt{H^3 T})$.

**Questions:**

(1) Before introducing Assumption 2, $\Phi(s,a), C(s'), \xi(s')$ are defined on line 184. If the author could explain these functions in more detail, it would be helpful to understand the Assumption 2.

(2) The current model parameter $\hat{W}_k$ is updated after the episode is over. However, since the current problem is a model-based setting, I think it is possible to update the model parameter on every horizon because the agent receives transition feedback on every horizon. If the agent can construct a confidence set for the true model parameter every horizon without considering the computation issue for planning, I think the regret bound might be tighter. I wonder what the author thinks about this.

minor typos
1. Line 566: I think it would be better to unify the numbering of Assumption 2 and the numbering in the appendix.
2. Line 579: $\hat{V}_n + \lambda I \succeq \lambda I$
3. Line 615: I think "Under Definition 1 and 2" should be fixed to "Under Definition 1 and Assumption 2"

**Limitations:**

I think there are no issues related to social impact. However, although this paper is highly related to theoretical part, considering that many recently published theoretical papers about model-based RL also present numerical experiments, I think it would be better if there is an experimental result in this paper.

**Strengths And Weaknesses:**

I think their research is relevant to RL community since they focus on how to design a provably and efficient algorithm for the nonlinear environment.
Beyond the existing work based on the linearity assumption for transition models or MDPs, as an extension of Exp-UCRL [10], which dealt with the problem when the transition models are parametrized by exponential families, the author proposes a more efficient method for estimating model parameters.
Based on the score matching technique, it can be more efficient than the previous method because it does not need to estimate the log-partition $Z_{sa}$.
Also, I think the analysis of the proposed algorithm is sound and the writing is clear.

However, in my opinion, the most important part of score matching relies on how to formulate the Fisher divergence (eq 3) into an empirical score matching loss (eq SM-L).
This result is presented in Theorem 1 and since this result is from [4], I carefully consider whether the result of this paper has significant novelty

Also, in Theorem 2, it presents the result of a self-normalized concentration guarantee when the parameter to be estimated $\hat{W}$ is vectorized.
I think this kind of result can be obtained if the problem setting $\exp(\langle \psi, W \phi \rangle)$ is replaced with $\exp(\langle vec(W), vec(\psi \phi^\top) \rangle)$.
If so, I think the bilinear structure disappears. Is there a way to express this concentration result without vectorization?

---

> ### Author Response · Authors · 2022-08-02
> **Thank you for your review.**
>
> We thank the reviewer for their comments and time.
>
> *To address the empirical performance*, we refer the reviewer to our response to Reviewer rkgm (so as not to be repetitive).
>
> Answering the questions below:
>
> 1. **Is there a way to express this concentration result without vectorization?** The short answer is we do not know how to do this, but we believe it is an interesting direction for future work. We make two comments.
>
>     a. We agree that a more general result would be to assume that the transition models follow the form $\exp(\langle \theta, \phi(s,a,s’)\rangle )$ where $\phi$ is some known feature mapping. This is a finite dimensional version of the exponential family introduced by [1]. However, our theoretical guarantees depend on the bilinear form, and it is unclear how to extend score matching to work in the general setting. The terms that appear in our score matching loss (see, e.g., Thm 1) depend on this bilinear structure. For example, in $\hat{V}_n$, we take derivatives with respect to the $\psi$ mapping.
>
>     b. Due to vectorization, we achieve a concentration guarantee for Frobenius norm. This occurs because our score matching estimator is solving ridge regression over the vectorized parameters. Note that the Exp-UCRL also obtains a Frobenius norm guarantee; while LC3 obtains a spectral norm guarantee (for nonlinear dynamical systems). It may be possible to prove a stronger regret guarantee for exponential family transitions that relies on spectral norm concentration.
> 2. **Explaining line 184 and Assumption 2.** To provide some intuition, it is helpful to compare to the standard setup of linear regression. Roughly speaking, one can view the $\Phi_t$ as covariates and the $\xi_t$ as the response. (Notice that $\Phi_t$ contains the information about the current $(s_t,a_t)$ pair, while $\xi_t$ contains information about the next state $s_{t+1}$.) However, score matching is different from linear regression because the “covariate matrix” $V_n$ contains an matrix $C$ which captures the “curvature” of the $\psi$ mapping.
> On a more technical level, in the proof of the concentration guarantee, we control the quantity $\hat{b}_n + \hat{V}_n \mathrm{vec}(W_0) = \sum \Phi_t (\xi_t + C_t W_0 \Phi_t)$. The term $\xi_t + C_t W_0 \Phi_t$ can be interpreted as the “error” term, which we assume is subgaussian in order to control (using (A) and (B)); however, in order to apply the self-normalized martingale guarantee we require (C) in order to change the matrix norm in Eq. 9. As for assumption (D), it is used to relate the KL divergence of two models to the distance between parameters in the regret analysis.
> In our revision, we will improve the discussion for these quantities.
> 3. **A tighter regret for updating during episode.** Yes, one might be able to achieve tighter regret guarantees if the concentration guarantee (Thm 2) is applied at every step in the episode. The downside is a higher computational burden. Both the score matching procedure and the planning procedure (which is hard to begin with and practically must be approximated) needs to be called $KH$ times instead of $H$ times.
>
> [1] Canu and Smola. “Kernel methods and the exponential family.”

---

> > ### Comment · Reviewer_LwWm · 2022-08-08
> > **Acknowledgement of Rebuttal**
> >
> > Thanks for your detailed response.
> > Given that my concerns have been addressed, I have updated my score.

---

### Official Review · Reviewer_rkgm · 2022-07-14

**Rating:** 7
**Confidence:** 3
**Soundness:** 3 good
**Presentation:** 3 good
**Contribution:** 3 good

**Summary:**

This paper presents a new model-based algorithm, called SMRL, for finite horizon episodic MDPs where the transition model is specified by exponential family distribution. Essentially, the work builds on Exp-UCRL [1] and proposes to use score matching instead of MLE to estimate the parameters of the transition model, which helps it run efficiently by eliminating the need to estimate the computationally expensive log partition function required in MLE. The proposed algorithm matches the regret bound offered by Exp-UCRL while being computationally efficient. The paper also presents theoretical proof for their efficient algorithm.

**Questions:**

Q1: How can we handle the problems for which the transition model does not belong to the exponential family distribution?
Q2: Can we approximate such transition models which do not belong to exponential family?

**Limitations:**

The main limitations is that the algorithm works on specific problems which satisfies the assumptions on transition and reward models, which limits the application of the algorithm.

**Strengths And Weaknesses:**

Strengths:
- Strong theoretical justification.
- The language of the paper is fine apart from some typos and uncommon abbreviations.

Weaknesses:
- The assumption that transition model belongs to exponential family may be limiting for many real-world problems.
- Although it is a theory paper, I would have liked to see how does the algorithm performs empirically vs Exp-UCRL.

---

> ### Author Response · Authors · 2022-08-02
> **Thank you for your review.**
>
> We thank the reviewer for their comments and time.
>
> **Regarding experimental results.** We are indeed interested in seeing how these ideas play out in practice.  We have successfully experimented with the estimation component, showing that score matching can indeed efficiently recover parameters for exponential family models - in particular, we can estimate models which *go beyond* nonlinear dynamical systems due to differences in the $q$ and $\psi$ functions, as we claim in the paper.  To see improvements in an actual RL problem, we need to combine this with a practical planning procedure (as with other approaches) and an interesting transition model, both of which require significant domain expertise.  We are now working with roboticists on stochastic control problems which can showcase the benefits of using SMRL (with a more expressive density) over the LC3 approach (i.e., fitting a nonlinear dynamical system).  This is a complex project, and in the meanwhile we hope that our ideas and theoretical methods, like other theoretical developments in RL, can inspire also other practitioners, as well as lead to further theoretical progress.
>
> Answering the (other) questions below:
>
> **What to do if the transition model doesn’t belong to the exponential family distribution? Can we approximate such transition models which do not belong to the exponential family?** Our theoretical results hold for the so-called “realizable” setting, where we assume the ground truth model lies in some model class (Definition 1). A more reasonable setting would be the “misspecified” or “agnostic” setting, where we assume that the transition model only approximates reality up to some “error”.
>
> Generally, understanding what the right notion of “error” is for model-based RL is a challenging open problem, not just for our setting. One example is the well-studied linear MDP [1,2]. The paper [2] shows that their algorithm LSVI-UCB adapts to misspecification in total variational distance to achieve $\mathrm{poly}(d,H,T)$ regret (see their Thm 3.2). However, if one weakens the notion of “error” to an $\ell_\infty$ notion of error, then the paper [3] establishes exponential lower bounds.
>
> We agree that it would be interesting to establish theoretical guarantees for SMRL which hold under misspecification, e.g., in TV distance. We leave this to future work.
>
> In practice, one can always run SMRL even in the presence of misspecification. In fact, control theorists have had tremendous success modeling complicated nonlinear systems as linear dynamical systems for decades! The appeal of modeling via exponential families (Definition 1) vs. just using linear dynamical systems is that (1) as a richer class, they can model more complicated densities (2) via score matching, they can still be estimated efficiently.
>
> [1] Yang and Wang. “Reinforcement learning in feature space: Matrix bandit, kernels, and regret bound.”
> [2] Jin, Yang, Wang, and Jordan. “Provably efficient reinforcement learning with linear function approximation.”
> [3] Du, Kakade, Wang, and Yang. “Is a good representation sufficient for sample efficient reinforcement learning?”.

---

> > ### Comment · Reviewer_rkgm · 2022-08-09
> > **Acknowledgement of Rebuttal**
> >
> > Thank you for the detailed response. It'll be interesting to see how this idea works out in practice. I'm satisfied with the response and have updated my score.

---

### Meta-Review · Area_Chair_bQsH · 2022-08-26

**Recommendation:** Accept
**Confidence:** Certain

**Metareview:**

This is a clear and carefully written paper with a solid mathematical contribution.  The reviewers are unanimous in supporting acceptance.

**Award:**

No

---

### Decision · Program_Chairs · 2022-09-14

Accept